# Benchmarking pipelines for subclonal deconvolution of bulk tumour sequencing data

Georgette Tanner [1], David R. Westhead [2], Alastair Droop [3] & Lucy F. Stead [1]✉

Intratumour heterogeneity provides tumours with the ability to adapt and acquire treatment resistance. The development of more effective and personalised treatments for cancers, therefore, requires accurate characterisation of the clonal architecture of tumours, enabling evolutionary dynamics to be tracked. Many methods exist for achieving this from bulk tumour sequencing data, involving identifying mutations and performing subclonal deconvolution, but there is a lack of systematic benchmarking to inform researchers on which are most accurate, and how dataset characteristics impact performance. To address this, we use the most comprehensive tumour genome simulation tool available for such purposes to create 80 bulk tumour whole exome sequencing datasets of differing depths, tumour complexities, and purities, and use these to benchmark subclonal deconvolution pipelines. We conclude that i) tumour complexity does not impact accuracy, ii) increasing either purity or purity-corrected sequencing depth improves accuracy, and iii) the optimal pipeline consists of Mutect2, FACETS and PyClone-VI. We have made our benchmarking datasets publicly available for future use.

[1] Leeds Institute of Medical Research, Faculty of Medicine and Health, University of Leeds, St James's University Hospital, Beckett Street, Leeds, West Yorkshire, LS9 7TF, UK. [2] School of Molecular and Cellular Biology, University of Leeds, Leeds, West Yorkshire, LS2 9JT, UK. [3] Wellcome Sanger Institute, Hinxton, Saffron Walden, CB10 1RQ, UK. ✉email: l.f.stead@leeds.ac.uk

Characterising genomic intratumour heterogeneity (ITH) is important for understanding how tumours evolve as cancers progress and in response to treatment. A common approach is to estimate the cancer cell fraction (CCF) of each somatic mutation from bulk sequencing data, and then attempt to delineate the tumour subclonal architecture by assigning those mutations into distinct clones. The different pipelines for performing such analyses have been found to give highly conflicting results[1–3] so robust benchmarking is needed to determine which to use. This, however, requires application of the methods to dataset(s) for which the subclonal architecture, the ground truth, is known. In cancer genomics studies, such benchmarking datasets are created via two approaches: (1) using real sequencing data in which mutation calls or CCFs have been ascertained through orthogonal approaches; or (2) using simulated datasets with a pre-defined subclonal architecture. The first approach is impacted by errors inherent to the orthogonal validation approaches used, and the second is dependent on how well the adopted simulation tools are able to recapitulate the complexity of tumour genomes and the noise inherent within sequencing data.

Previous subclonal deconvolution benchmarking studies have used a mixture of both approaches. The ICGC-TCGA DREAM Somatic Mutation Calling - Tumour Heterogeneity Challenge used a wrap-around script for BAMSurgeon, a programme that inserts mutations directly into aligned sequencing reads from a real high coverage input dataset and then uses down-sampling to simulate copy number variation[2,4]. This, however, bypasses the challenges that mutation-containing reads pose to the alignment stage itself, as well as the inability to ascertain all true positive mutations in any real data with complete certainty. An alternative approach that was taken by the Pan-Cancer Analysis of Whole Genomes Evolution and Heterogeneity Working Group was to simulate mutation call sets that fit given ground truths, with errors added from simple distributions to represent imperfect mutation calling[5–7]. However, this negates assessment of the different callers used to generate the mutation inputs and is unlikely to fully capture the inaccuracies seen in real mutation call sets. In addition, assumptions have been made to simplify the simulation of call sets, meaning the complexity of real tumours is not fully modelled. Such assumptions include not modelling CNVs[6], subclonal mutations only able to lie on one copy of a chromosome[5], or no more than two copy number states per region. Other studies have instead used real data to draw comparisons between pipelines, allowing for biases to be identified but with no certainty of the ground truths[3].

In this study, we benchmark pipelines for subclonal deconvolution, including methods for inferring and clustering of mutation CCFs, as well as those needed upstream to produce the inputs; mutation and copy number callers. We use a state-of-the-art tumour genome simulation tool, HeteroGenesis, that we developed to overcome issues identified with previous simulation tools[8]. We then apply w-Wessim2, which builds upon w-Wessim[8,9] to allow in silico whole-exome sequencing (WES) of simulated genomes. w-Wessim2 produces reads with alignment distributions that realistically mirror those of real datasets and incorporates the sophisticated sequencing error models of the in silico sequencer ReSeq to recapitulate errors in real data[10].

Figure 1 shows the test datasets that were created for our study. Three parameter sets (S1-3) were used to produce tumours with increasing point mutation rates and copy number variations. Three replicate genomes (R1-3) were simulated per parameter set. Tumours were in silico sequenced to different sequencing depths (30x, 60x, 100x and 250x). A corresponding germline genome was simulated, per tumour, and in silico sequenced to 60x. This was also used to incorporate normal cell infiltration to assess the effect of tumour purity across a range of 25–100%. Figure 2 outlines the process for test set generation and benchmarking of four mutation callers (Mutect2[11], Strelka2[12], VarScan2[13] and Lancet[14]), four copy number callers (Sclust[15], FACETS[16], Sequenza[17] and TITAN[18]) and five subclonal deconvolution tools (PyClone[19], PyClone-VI[7], FastClone[20], Ccube[21] and Sclust[15]).

We have, therefore, performed the most comprehensive, robust and accurate assessment of subclonal deconvolution analysis pipelines to date and have made our series of highly realistic tumour WES datasets, spanning a range of biological and dataset-driven parameters, available to the wider community for future benchmarking studies as new tools emerge.

## Results

**Somatic point mutation calling suggests Mutect2 is best for subclonal variants.** We applied four somatic variant callers (Mutect2[11], Strelka2[12], VarScan2[13] and Lancet[14]) to a highly mutated tumour dataset (S3R3) at 100% tumour purity across all sequencing depths. Runtimes differed substantially between methods, with Strelka2 being fastest and Mutect2 taking longest (Supplemental Fig. 1). Performance was assessed on both unfiltered and filtered (an additional step attempting to triage mutation calls based on the probability of being a technical artifact, non-somatic or a sequencing error) call sets. Precision-recall curves, using mutation probabilities or scores in thresholding, show low recall rates for all callers, as expected due to the low cellular frequency of many subclones (Fig. 3a). Nonetheless, most mutations with a variant allele frequency (VAF) ≥0.1 were detected by most callers at >100x coverage; the notable exception being VarScan2 (Fig. 3b). Strelka2_filtered achieved the highest F1 score (which conveys the balance between precision and recall) of 0.526–0.738, followed closely by Mutect2_filtered (0.484–0.723), while VarScan2 scored lowest of the filtered call sets (0.246–0.319). The high precisions (0.882–0.896) achieved by Strelka2_filtered pertained to calls within target exon regions, but precision dropped substantially (0.360–0.660) when expanding to calls across the whole genome, suggesting a high false positive call rate in lower coverage regions. The other callers showed similar precisions across exons and the whole genome (Supplemental Tables 1 and 2). Two additional call sets consisting of the union and intersect of calls from the best achieving variant callers, Mutect2_filtered and Strelka2_filtered, were analysed to investigate the benefits of ensemble approaches. Union call sets had slightly improved F1 scores owing to small increases in recall with minimal loss of precision. Intersect call sets, however, had reduced F1 scores resulting from reduced recall (Fig. 3a, Supplemental Table 1). To investigate potential differences between WES and whole-genome sequencing (WGS), we created in silico WGS reads for sample S2R2_B_100x and compared variant calling performance from this with WES for the same sample. Of note, Strelka2_filtered dropped in both precision and recall for WGS compared to WES, whereas Mutect2_filtered maintained a similar performance (Supplemental Table 3).

**Subclonal somatic copy number calls are most accurately quantified by FACETS.** When used as input for subclonal deconvolution, somatic copy number alteration (CNA) calls must accurately denote regions of chromosomal gain or loss but also the allelic copy numbers. We therefore assessed CNA callers in three ways: (1) purity and ploidy estimates were assessed through direct comparison to true values; (2) heatmaps of predicted and true total and allelic copy numbers were generated to assess the relative performance of methods and the factors affecting them; (3) results from all the CNA callers were used with subclonal deconvolution methods, and the accuracy of the resulting CCF estimates were compared.

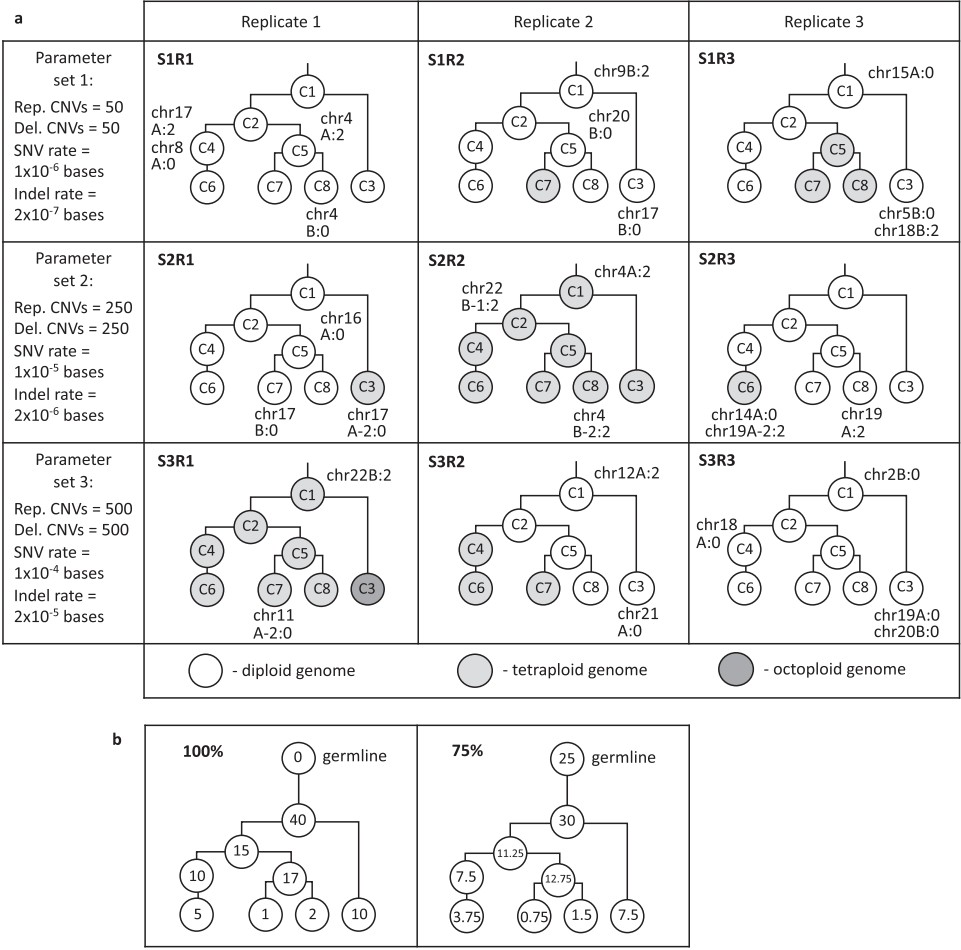

**Fig. 1 The ground truth datasets. a** Tumour clonal architectures created by HeteroGenesis. Clones are denoted C1 to C8. Single chromosome aneuploid events are indicated next to the clone in which they first appear, but are also present in all daughter clones. Three different parameter sets (S1 to S3) were used to define total somatic mutation numbers in each tumour, with three replicates created for each set (R1 to R3). **b** Clone and germline proportions represented in the bulk samples, with numbers in circles representing the relative proportions of reads from each, for 100 and 75% tumour purities. 50 and 25% samples were similarly created for a subset of tumours. CNV copy number variation, chr chromosome, Rep replication, Del deletion, SNV single-nucleotide variation.

We applied four somatic allele-specific CNA callers (Sclust[15], FACETS[16], Sequenza[17] and TITAN[18]) to samples from all nine simulated tumours at 100 and 75% purity and across sequencing depths of 30x–250x. We further included 50 and 25% purity samples at 250x for three of the tumours. Ploidy and purity estimates were largely accurate, particularly at ≥50% purity, with the exceptions that TITAN often incorrectly assessed purity and Sequenza frequently overestimated ploidy (Fig. 4, Supplemental Table 3). None of the callers were able to reliably detect whole-genome duplications, and instead estimated them as being diploid. This is to be expected as all approaches must use relative depth ratios across the genome to predict more focal CNAs, making whole-genome duplications almost undetectable. Sequencing depth was not found to affect ploidy or purity estimates (Fig. 4, Supplemental Table 3).

Heatmaps of gains and losses along the genome for predicted and true copy numbers also indicated that sequencing depth had little effect on performance, with the majority of CNAs called at 250x also called at 30x (Supplemental Fig. 2). An exception was with Sequenza, which falsely called multiple loss of heterozygosity regions at 250x only, as a result of over-segmentation. Tumour purity also had a limited effect on results. All methods, and particularly Sclust, failed to detect CNAs present in low-frequency subclones, where the total copy number remains close

to two, although FACETS was most sensitive. TITAN frequently falsely called large regions of single allele whole-genome gains or losses, particularly in the S1 samples, which have the lowest variant frequencies. None of the methods were able to detect subclonal whole-genome duplications with the exception of Sequenza which, when it did, identified them as clonal.

**CCF estimation is most accurate at higher sequencing depth and following clustering.** The performance of five subclonal deconvolution methods (PyClone[19], PyClone-VI[7], FastClone[20], Ccube[21] and Sclust[15]) was assessed on all tumours and sequencing depths, at 100 and 75% purity. We used the Mutect2_filtered variant calls combined with CNA calls, purity and ploidy estimates from all four CNA callers as inputs for subclonal deconvolution (Fig. 2). PyClone, PyClone-VI, FastClone and Ccube were run with CNA inputs from Sequenza, FACETS and TITAN, whereas Sclust was run using only its own CNA calls. PyClone-VI performs clustering using either binomial or beta-binomial distributions so we ran it with both separately. PyClone was unable to complete within 48 h (the time limit on our high-performance computing system) for most of the highly mutated S3 samples and these were, therefore, omitted. FastClone did not converge for 61 of the 216 runs so these were also not included. The

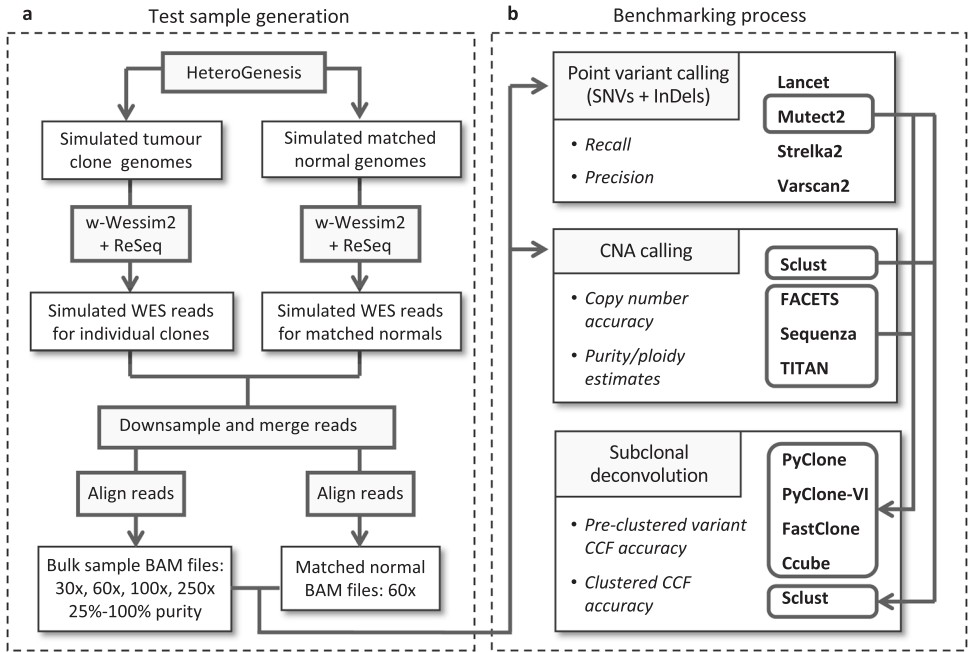

**Fig. 2 Test sample generation and benchmarking processes. a** Replicate tumour, and matched normal, genomes were simulated using HeteroGenesis according to the parameter sets outline in Fig. 1. In silico WES data was created with w-Wessim2, with errors incorporated via ReSeq. Reads were downsampled and merged to create bulk tumour sequencing data with varying purity and mean sequencing depths, before being aligned. **b** Bulk tumour and matched normal alignments were used as input for somatic point mutation and copy number callers. Calls from the top performing point mutation caller, Mutect2, were combined with output from all four copy number callers and input to five subclonal deconvolution methods. Sclust performs both copy number calling and subclonal deconvolution and is therefore not tested with outputs from other copy number callers. The metrics used to assess the performance of each algorithm is in italics. CNA copy number alteration, SNV single-nucleotide variation, CCF cancer cell fraction.

performance of each pipeline was determined through comparison of true variant CCFs (calculated by summing the tumour proportions of clones that contained each variant) to both the estimated non-clustered variant CCFs and the CCF of clusters that the variants were subsequently assigned to, where applicable. This enabled inspection of whether accuracy is impacted primarily at VAF normalisation or during clustering.

To provide additional simplistic references upon which CCF estimation methods should improve, we developed baseline datasets via two approaches. For the non-clustered CCFs, "doubled VAFs" was implemented in which CCFs were estimated by simply doubling VAFs, dividing by true purity (which was generally accurately estimated by all CNA calling methods) and limiting to ≤1 i.e. assuming a purely heterozygous diploid genome. As a baseline for comparing the clustered CCFs against, we applied K-means clustering ($n = 3$) to the doubled VAFs.

The mean absolute differences (MADifs) between true and predicted CCFs are shown in Fig. 5 for both non-clustered and clustered CCF estimates with low values indicating better accuracy. In addition, mean absolute adjusted Rand index is calculated to indicate the accuracy of the variant clustering in each pipeline (Supplemental Table 5). When using non-clustered CCF estimates the best performing pipeline is FACETS with PyClone, though this only marginally improves upon the baseline of doubled VAFs and only at sequencing depths ≥100x (Fig. 5a). When using clustered CCF estimates, the MADif and adjusted Rand index metrics largely agreed, with the best performing pipeline being FACETS with PyClone-VI run with beta binomial used for clustering. The results do show that clustering CCFs improves estimates, and most pipelines improve upon the simplistic baseline, created by K-means clustering of doubled VAFs, at all sequencing depths but, again, only marginally (Fig. 5b, Supplemental Table 5).

The performance metrics did not vary much between pipelines using different CCF estimation methods, with the exception of FastClone which performed worse than any of the others. In contrast, those using FACETS as CNA caller consistently showed improved accuracy and those using TITAN performed the worst. Both non-clustered and clustered CCF estimates increased in accuracy with increasing sequencing depth, whereas, varying the frequency of mutations (which increased from S1 to S3 tumours) did not show a consistent influence (Supplemental Fig. 3).

We next sought to investigate the effects of lower tumour purities on the top three performing tumours (S1R1, S2R1, S2R3), using samples at 100, 75, 50 and 25% purity. This showed a decrease in accuracy with decreasing purity, particularly at 25% (Supplemental Fig. 2), which likely partially results from the poorer purity estimates from CNA callers at that purity. To further assess the robustness of our findings, we ran the pipelines on additional WES datasets for samples with reduced numbers of clones and altered clone topologies, as well as the WGS datasets for the top (S1R1) and worst (S2R2) performing WES samples. In all investigations, the relative performance of pipelines was largely preserved, with FACETS with PyClone-VI_beta-binomial remaining the top performers (Supplemental Figs. 4–6).

## Discussion

We have benchmarked the performance of methods used in subclonal deconvolution from bulk single tumour WES and matched normal samples. This is an important goal, as subclonal deconvolution pipelines have not been extensively benchmarked previously, and the few studies that have attempted it include approaches that may introduce certain biases[2,5]. We therefore aimed to expand upon previous studies by recapitulating both the complexity of tumour genomes and the noise introduced during

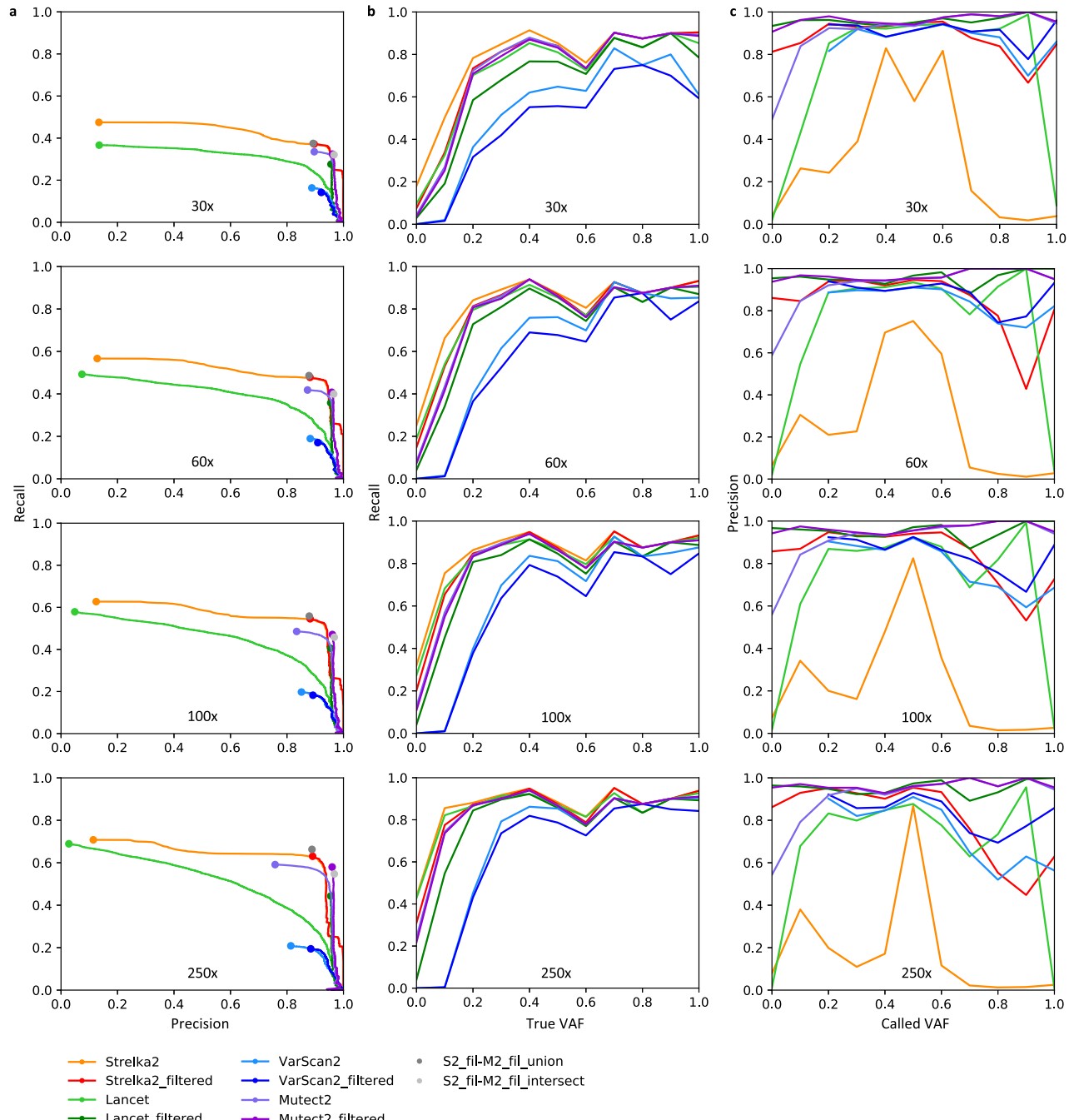

**Fig. 3 Performance of variant callers applied to sequencing data from tumour S3R3 with 100% purity at 30x, 60x, 100x and 250x coverage.** Each variant calling method was assessed across target exon regions, both pre- and post-filtering (denoted with _filtered suffix). S2_fil-M2_fil_union is the call set resulting from the union of results from Strelka2_filtered and Mutect2_filtered, with the intersect of results from those callers denoted by S2_fil-M2_fil_intersect. **a** Precisions–recall curves. **b** Recall according to true variant allele frequency (VAF). **c** Precision according to called variant allele frequency. Source data are provided as a Source data file.

whole-exome sequencing (WES) when simulating data on which to test subclonal deconvolution pipelines. We did this through the use of HeteroGenesis[8], w-Wessim2, and ReSeq[10], which together allow for realistically complex tumour WES dataset generation. This enabled us to carry out reliable assessment of all steps from unprocessed reads through somatic point and copy number mutation calling and through to CCF inference and clustering. We did not attempt to assess methods for phylogenetic reconstruction as often multiple trees are outputted requiring manual

curation, and which are not possible to categorically compare. Nonetheless, the accuracy of phylogenies will be heavily impacted by that of the inputted mutation CCF estimates.

Our results indicate that Mutect2 is the best performing somatic point mutation caller. Ensemble callers, which combine results from multiple programmes, can be used to generate high confidence call sets[22–24]. We showed, by taking the union and intersect of the two best performing individual callers in our study, that such methods risk increasing errors or decreasing the

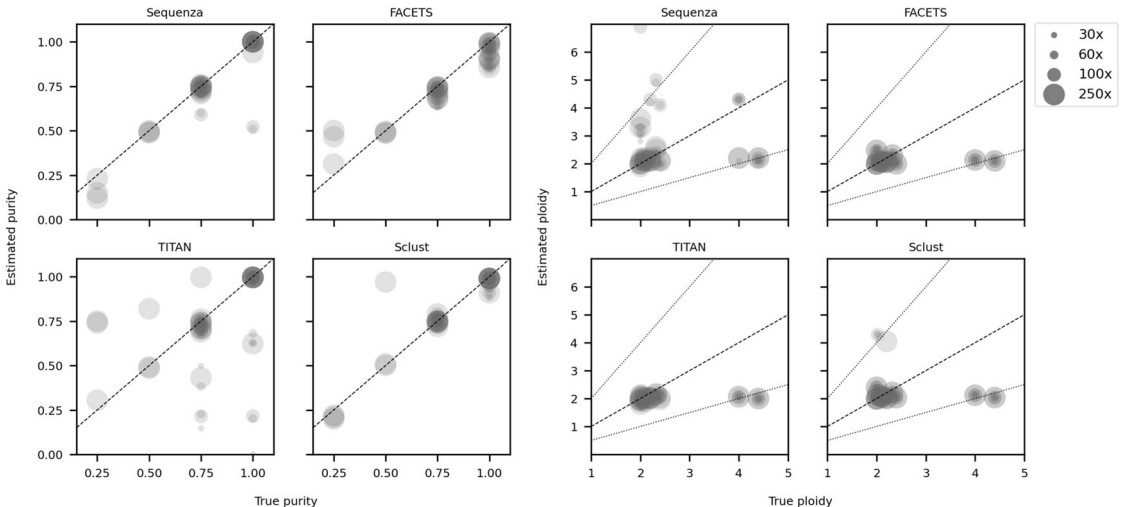

**Fig. 4 Accuracy of purity and ploidy estimates from tested somatic copy number alteration (CNA) callers.** Dashed lines denote accurate ploidy and purity estimates, and dotted lines indicate ploidy estimates where the caller misclassed samples as tetraploid instead of diploid, or vice versa, but were otherwise accurate. Source data are provided as a Source data file.

number of true positives compared to using single methods alone. This could be especially detrimental when poorer performing callers are included.

We quantified the accuracy with which pipelines estimate both non-clustered and clustered mutation CCFs. While the choice of CCF estimation method had an effect on this, our results suggest the choice of CNA caller is equally important. FACETS was found to be more sensitive in detecting CNAs than the other methods, resulting in improved accuracy for CCF estimation.

PyClone is possibly the most widely used subclonal deconvolution method but was designed for targeted sequencing and suffers from lack of scalability to higher variant numbers. We found that PyClone-VI was the best performing CCF estimation tool but that Ccube performed similarly well, in agreement with a recent study by Dentro et al.[5]. However, in contrast to the DREAM study[25] where FastClone performed well, it was found to be least accurate when applied to our range of tumours with different underlying biological and technical parameters.

Our results indicate that a sequencing depth of 250x is superior to lower depths for calling subclonal point mutations present in fewer than 10% of cells, in agreement with our previous work[26], and also enables the most accurate CCF estimates, but this should be corrected for tumour purity where possible.

An important feature of tumour clones is that they continue to develop new mutations as the cells divide and grow. This results in a neutral tail of low VAFs, where the cumulative number of variants has a linear relationship with the inverse of their allelic frequency[27]. The neutral tail is distinct from the major peak of VAFs present in every cell of a clone and with most CCF clustering methods, it may wrongly be identified as primary peaks from additional clones. MOBSTER is a programme that applies evolutionary modelling to non-clustered CCFs in order to identify and remove neutral tails during subclonal deconvolution, and allow clusters of true clones to be determined more accurately[28]. Neutral tails were not modelled in the datasets created in this study, hence we did not apply MOBSTER but this may be useful for real data after generating non-clustered CCFs using one of the CCF estimation methods tested in this study.

Regardless of the computational tools used, somatic point and copy number mutation calling, and subsequent subclonal deconvolution are ultimately limited by current sequencing technologies and depth[29]. In future, single-cell analyses may overcome many of the issues with investigating intratumour heterogeneity. Currently, these methods are expensive, produce a substantial amount of noise due to drop-outs and are technically challenging to apply to archived material[30,31]. Our benchmarking of steps involved in subclonal deconvolution is, therefore, necessary to inform researchers on the most suitable methods to apply to bulk tumour sequencing data, which is still the most pragmatic option for most studies. We have made our benchmarking datasets freely available at https://www.ebi.ac.uk/ena/browser/view/PRJEB28319 and https://github.com/GeorgetteTanner/benchmarking to facilitate assessment of additional methods as they emerge.

## Methods

**Dataset simulation**. HeteroGenesis (v1.5) was used to simulate nine sets of tumour genomes from three parameter sets, indicated in Fig. 1. Other parameters common to all tumours included: "snvgermline":0.0014, "indgermline":0.00014, "aneuploid":4, "wgdprob":0.333, "cnvrepgermline":160, "cnvdelgermline":1000, "cnvgermlinemean":-10, "cnvgermlinevariance":3, "cnvgermlinemultiply":1000000, "cnvsomaticmean":-1, "cnvsomaticvariance":3, "cnvsomaticmultiply":1000000, "indmean":-2, "indvariance":2, "indmultiply":1, "cnvcopiesmean":1, "cnvcopiesvariance":0.5, "dbsnpsnvproportion":0.9, "dbsnpindelproportion":0.5, "chromosomes":["all"], "structure":"-clone1,2,germline,clone2,1,clone1,clone3,3,clone1,clone4,1,clone2,clone5,1,clone2,-clone6,1,clone4,clone7,1,clone5,clone8,1,clone5".

The human reference genome hg38 (chromosomes 1–22) formed the basis of simulations. Germline variants were taken from dbsnp_146.hg38.vcf. A basic local alignment tool (BLAT) alignment[32] (pBlat[33]) of real WES data (NCBI Sequence Read Archive, accession no. SRR2103613, cleaned and filtered for a high mapping score) was provided to w-Wessim2 to define regions for sequencing. Simulated genome sequences for each germline and subclone in the tumours were in silico whole-exome sequenced to produce 151 bp paired-end reads using w-Wessim2 (v1.0) and ReSeq (v1.1) to ~134x mean sequencing depth across target exon regions. Sequencing errors were incorporated through the seqToIllumina module of ReSeq, using an error profile for NovaSeq 6000 (Hs-Nova-TruSeq.reseq from https://github.com/schmeing/ReSeq-profiles). WGS reads were created using Reseq's illuminaPE command. Resulting fastq files were down-sampled and merged to create heterogeneous bulk samples at required depths, as indicated in Fig. 2. Sequencing depths were determined from the mean target coverage identified by Picard CollectHsMetrics (v2.19.1-SNAPSHOT-all) (http://broadinstitute.github.io/picard/). Reads were cleaned with cutadapt (using the parameters: -a AGATCGGAAGAGC -A AGATCGGAAGAGC -m 20 -O 1 -q 20) and aligned to the with BWA-MEM[34].

**Point mutation calling**. Mutect2 (GATK v4.1.7.0) was run under default parameters, with af-only-gnomad.hg38.vcf.gz provided as the germline_resource. FilterMutectCalls was run under default parameters. Strelka2 (v2.9.10) was run under default parameters with the addition of the "--exome" flag (for WES data only) and

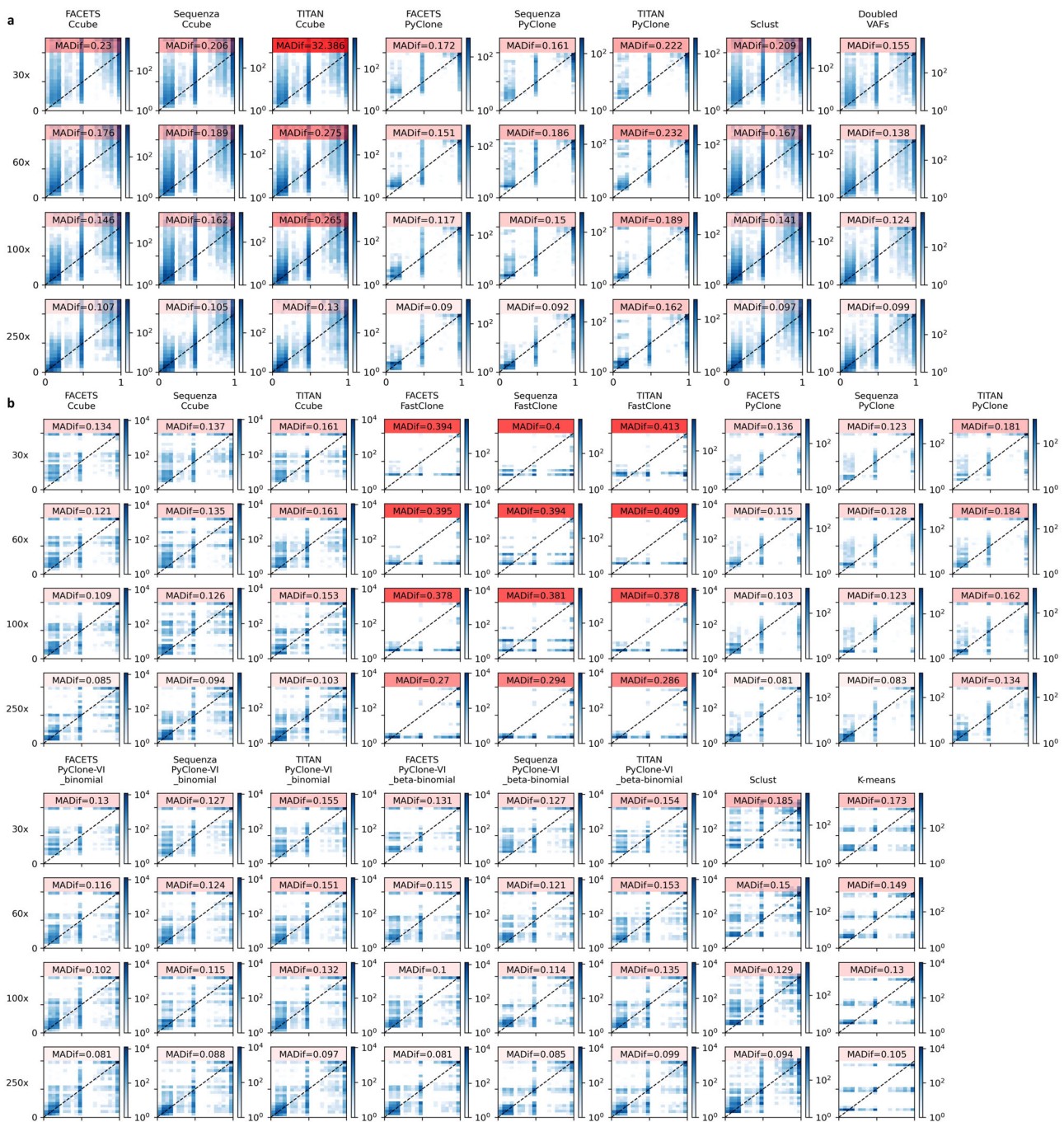

**Fig. 5 True CCF (*x*-axis) versus predicted CCF (*y*-axis) from subclonal deconvolution pipelines applied to tumours sequenced to different depths.** Plots are separated by predictions based on **a** non-clustered CCFs, for which 'Doubled VAFs' represents the simplistic baseline from which to assess performance; and **b** clustered CCFs, for which K-means provides the simplistic baseline. $N = 18$ samples at each depth. MADif: weighted mean absolute difference of true vs. predicted CCFs with the depths of colour corresponding to the value. CCF cancer cell fraction, VAF variant allele frequency. Source data are provided as a Source data file.

the full lengths of chromosomes 1–22, X, Y, M provided to "--callRegions". Lancet (v1.1) was run under default parameters. VarScan2 (v2.4.4) was run with the flag "--dream3-settings 1", on samples that had been realigned in pairs of tumour-normals by GATK (v3.8-1-0)[35] and converted to mpileup format with samtools (v1.9). VarScan2's fpfilter was run using the input from bam-readcount (v0.8.0) and with the flags "--dream3-settings 1 --keep-failures 1". All alleles in the bam-readcounts output were converted to uppercase to avoid an error we encountered with lower case allele not being recognised by fpfilter. The scikit-learn package was used to create precision-recall curves (v0.23.1)[36]. Targeted exon regions are defined as the Agilent SureSelect All Exon v5+UTR kit covered regions. WGS variant calls were limited to those outside of blacklist regions, from https://github.com/Boyle-Lab/Blacklist/raw/master/lists/hg38-blacklist.v2.bed.gz.

**CNA calling.** Sequenza (v3.0.0, sequenza-utils v3.0.0) was run with the provided pype pipeline. All default settings were used, including a binning size of 50. TITAN (v1.23.1, ichorCNA v0.3.2, HMMcopy v1.26.0) was run using the provided sna-kemake workflow with default parameters, with the following exceptions: parameters were set for use with the hg38 human reference genome; a BED file of regions covered by the S04380219 Agilent SureSelect All Exon v5+UTR probes was provided to define target regions (ichorCNA_exons); chromosome X was removed from the analysis; initial normal contamination values were set to "c(0,0.1,0.2,0.3,0.4,0.5,0.6)"; the maximum number of clusters was set to 8; a statistical parameter was adjusted to accommodate WES datasets (TitanCNA_alphaK: 2500). FACETS (v0.5.14) was run using the wrap around script, cnv_facets (v0.15.0) (https://github.com/dariober/cnv_facets). Default parameters were used,

with the exception of automatically detecting an appropriate insert size with "--nbhd-snp auto". A bed file specifying target exon regions was provided. CNA cellular fraction taken from the CF_EM field. Sclust (v1.1) was run as described in Cun et al.[15], with "-indel" added to the cluster module to include indels. Parameters for CNA callers were adjusted where appropriate for use with WGS datasets, including adding "--cval 25 400" for FACETS. Sclust required the addition of "-lambda 0.0000002" to complete on WGS_S1R1_100x. Heatmaps were generated using a customised version of CNVkit in which three script files were altered (altered files are present in the GitHub repository associated with this paper) to enable bespoke figure formatting[37].

**CCF estimation**. Purity estimates for all subclonal deconvolution methods were taken from the same CNA caller used to provide copy number inputs. PyClone (v0.13.1) was run with the addition of '--init_method connected --num_iters 100000' as recommended when more than a few hundred variants are present. PyClone-VI (v0.1.0) was run with a maximum of 10 clusters and 100 random restarts, with both binomial and beta-binomial probability densities separately. When PyClone, PyClone-VI or FastClone were used with outputs from FACETS and TITAN, only CNAs estimated to be in ≥50% cells were incorporated into the inputs. Ccube (v1.0) was run with the command, 'RunCcubePipeline(ssm = data, numOfClusterPool = 1:10, numOfRepeat = 1, runAnalysis = T, runQC = T, multiCore = F)' as indicated in https://github.com/keyuan/ccube/blob/master/inst/test_subclonal_cn_snv.R, and providing either clonal copy number estimates from Sequenza or clonal and subclonal estimates from TITAN/FACETS. Sclust details are described above with CNA callers.

K-means clustering was performed with scikit-learn (v0.23.1)[36]. MADif metrics for comparing true vs. predicted CCFs were calculated with each sample weighted equally so that those with the highest variant numbers did not dominate the results:

$$MADif = \frac{1}{n}\sum_{i=1}^{n}\frac{\sum_{j=1}^{m}|T_{S_{ij}}-E_{S_{ij}}|}{m}$$

where $n$ is the number of samples ($S$) in a group, m is the number of called true variants, and $T$ and $E$ are the true and estimated CCF values for a variant.

**Reporting summary**. Further information on research design is available in the Nature Research Reporting Summary linked to this article.

## Data availability
The simulated sequencing data generated in this study have been deposited in the European Nucleotide Archive database under accession code PRJEB28319. The simulated sample and clone mutation profiles and HeteroGenesis inputs generated in this study have been deposited on GitHub [https://github.com/GeorgetteTanner/benchmarking]. Source data are provided with this paper.

## Code availability
The code used to generate the datasets in this study is publicly available at https://github.com/GeorgetteTanner/HeteroGenesis[38] and https://github.com/GeorgetteTanner/w-Wessim2 [39].

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

## Acknowledgements
We would like to acknowledge S. Schmeing, the author of ReSeq, for his support in using that software. This work was undertaken on ARC3, part of the High Performance Computing facilities at the University of Leeds, UK. This work was made possible owing to funding from UK Research and Innovation (UKRI) grant number MR/T020504/1 (L.F.S.).

## Author contributions

G.T. aided in design of the work, created the software used and acquired, analysed and interpreted the data and drafted the manuscript. D.R.W. aided in the design of the work and reviewed the manuscript. A.D. aided in the design of the work and the analysis of data and reviewed the manuscript. L.F.S. conceived the work, aided in the design, supervised the data analysis and substantially revised the manuscript.

## Competing interests

The authors declare no competing interests.
