## [Peer Review File · Nature Communications]

REVIEWER COMMENTS

Reviewer #1 (Remarks to the Author): Expert in intratumour heterogeneity, clonal deconvolution methods, and bioinformatics

This paper benchmarks several methods for deconvolution of bulk DNA-seq data of tumors. Specifically, the paper focuses on methods for SNV calling, CNA calling and SNV clustering. The paper is well written. I do have some comments regarding the novelty of the analysis.

1. Not the first benchmarking study

Paper states:

"We have, therefore, performed the most comprehensive, robust and accurate assessment of subclonal deconvolution analysis pipelines to date and have made our series of highly realistic tumour WES datasets, spanning a range of biological and dataset-driven parameters, available to the wider community for future benchmarking studies as new tools emerge."

A similar study appeared recently, also in Nature Comm:

Liu, L. Y., Bhandari, V., Salcedo, A., Espiritu, S. M. G., Morris, Q. D., Kislinger, T., & Boutros, P. C. (2020). Quantifying the influence of mutation detection on tumour subclonal reconstruction. *Nature Communications*, 11(1), 6247. <https://doi.org/10.1038/s41467-020-20055-w>

This decreases the novelty of the paper.

2. Simulations

Abstract states: "... the most comprehensive tumour genome simulation tool available ...".

There is another tool for doing this: MASCoTE, published in:

Zaccaria, S., & Raphael, B. J. (2020). Accurate quantification of copy-number aberrations and whole-genome duplications in multi-sample tumor sequencing data. *Nature Communications*, 11(1), 4301. <https://doi.org/10.1038/s41467-020-17967-y>

In addition, it would be good to vary the tree topology as well as the number of clones. Also, varying the number of samples would be good to do.

3. Assess mutation clustering

In addition to comparing inferred CCF to ground-truth CCF, it would be good to compare mutation clustering accuracy using V-measure or (adjusted) Rand index.

Minor

* Page 3: "This was also used to incorporate immune cell infiltration to assess the effect of tumour purity at 75% vs 100%."

Immune cells are not the only type of normal cells.

* Figure 3: Is panel (b) showing recall split by VAF? If so, change y-label to "Recall" as in panel (a). Also, it would be good to show precision split by VAF

* SNV clusters are not clones

Methods for SNV clustering do not infer tumor clones. Tumor clones can only be inferred in the context of a phylogenetic tree.

Reviewer #2 (Remarks to the Author): Expert in cancer genomics and evolution, clonal deconvolution, and bioinformatics

This is a study that benchmarks different tools for single nucleotide variant (SNV) calling, copy number aberration (CNA) analysis and subclonal deconvolution with the aim of finding the best combination to study tumor evolution from bulk whole-exome sequencing (WES) data. The authors first used their published tool to simulate tumor genomes with different subclonal structure as well as SNV and CNA rates. They then used another tool to generate whole exome sequencing data at a range of sequencing coverage (30x, 60x, 100x and 250x) and tumor purity (75% and 100%). This data is used to find the best combination from a set of four SNV callers, four CNA callers and five subclonal deconvolution tools. Their analysis indicates that Mutect2/FACETS/PyClone-VI combination gives the optimal solution. This is an interesting study but there are some important points that need to be addressed:

1. The choice of WES as the sequencing data is intriguing as there might not be enough mutations to cluster unless the tumor is a hypermutant (~10 mutations/Mb). A recent study showed that most studied tumors do not have a high mutation burden [Campbell et al Cell 2018]. 17% of the tumors had 10Muts/Mb and only 0.6% had 100Muts/Mb. These correspond to parameter set-2 and -3 in Figure 1. This means that only ~33% of their input data will reflect the mutation burden observed in the majority of tumors. This could be exacerbated by using whole genome data. Did the authors check how the choice of WGS and WES would affect their results?
2. There are only two purity values taken into account: 75% and 100%. The authors should include lower purity values as 75% is still very high for most solid tumors.
3. I am a bit surprised that Mutect2 came out as the caller of choice. Our experience with Mutect2 using WGS is that it has a higher false positive rate compared to Strelka2 especially in the low VAF range. That is the reason why we always prefer ensemble calling to using a single mutation caller. But the difference may be because of the use of WES in this study.
4. The authors assessed the performance of subclonal deconvolution tools comparing 'true variant CCFs to the estimated non-clustered CCFs and the CCF of the clusters the variants are assigned to'. True variant CCFs are calculated as multiplying the VAF by two and dividing by purity. This does not take CN changes into account. Would the results be the same if the authors excluded mutations in genomic regions affected by CNA?

Reviewer #1 (Remarks to the Author): Expert in intratumour heterogeneity, clonal deconvolution methods, and bioinformatics

This paper benchmarks several methods for deconvolution of bulk DNA-seq data of tumors. Specifically, the paper focuses on methods for SNV calling, CNA calling and SNV clustering. The paper is well written. I do have some comments regarding the novelty of the analysis.

1. Not the first benchmarking study

Paper states:

"We have, therefore, performed the most comprehensive, robust and accurate assessment of subclonal deconvolution analysis pipelines to date and have made our series of highly realistic tumour WES datasets, spanning a range of biological and dataset-driven parameters, available to the wider community for future benchmarking studies as new tools emerge."

A similar study appeared recently, also in Nature Comm:

Liu, L. Y., Bhandari, V., Salcedo, A., Espiritu, S. M. G., Morris, Q. D., Kislinger, T., & Boutros, P. C. (2020). Quantifying the influence of mutation detection on tumour subclonal reconstruction. Nature Communications, 11(1), 6247. <https://doi.org/10.1038/s41467-020-20055-w>

This decreases the novelty of the paper.

We thank the reviewer for pointing out this study which we have now mentioned in our introduction. Unlike our approach, Liu et al. do not compare pipeline outputs to known ground truths and instead compare pipelines run on real data, allowing for interesting observations on biases but without estimation of the overall accuracy of methods. There was also a need for assessment of more up to date variant callers (eg. Liu et al., 2020 use Mutect1 v1.1.4, whereas the latest version is Mutect2 v4.2.0). We therefore believe our study compliments results from the Liu et al. study but has important distinctions that allow it to be considered novel.

Line 107: "Other studies have used real data to draw comparisons between pipelines, allowing for biases to be identified but with no certainty of the ground truths³."

2. Simulations

Abstract states: "... the most comprehensive tumour genome simulation tool available ...".

There is another tool for doing this: MASCoTE, published in:

Zaccaria, S., & Raphael, B. J. (2020). Accurate quantification of copy-number aberrations and whole-genome duplications in multi-sample tumor sequencing data. Nature Communications, 11(1), 4301. <https://doi.org/10.1038/s41467-020-17967-y>

We also thank the reviewer for pointing out this study. MASCoTE simulates copy number alterations but does not include simulation of SNVs or small InDels and is therefore not suited to the analyses of the current study. Given the tight word limit of the manuscript it is not possible to go into details of individual alternative simulation tools but we hope our slight change to the wording better implies that HeteroGenesis is the most comprehensive out of those suitable for the purposes of our study.

Line 16: "... but there is a lack of systematic benchmarking to inform researchers on which are most accurate, and how dataset characteristics impact performance. To address this, we used the most comprehensive tumour genome simulation tool available **for such purposes** to create 80 bulk tumour whole exome sequencing datasets of differing depths, tumour complexities, and purities, and used these to benchmark subclonal deconvolution pipelines."

In addition, it would be good to vary the tree topology as well as the number of clones. Also, varying the number of samples would be good to do.

We agree that the study would benefit from additional variations of the samples. To address this we created additional samples with 3 and 5 clones, and which have a linear or branched tree topology respectively, and ran the pipelines on these. The relative performance of the different pipelines was largely unchanged and did not alter our conclusions.

Line 332:" To further assess the robustness of our findings, we ran the pipelines on additional WES datasets for samples with reduced numbers of clones and altered clone topologies, as well as the WGS datasets for the top (S1R1) and worst (S2R2) performing WES samples. In all investigations, the relative performance of pipelines was largely preserved, with FACETS with PyClone-VI_beta-binomial remaining the top performers (Supplemental Figures 4-6)."

The suggestion of varying the number of samples is also of interest. However we believe a multisample set-up to be a distinct scenario (for example it would benefit from including variant and copy number callers specifically designed to be used with multiple samples such as HATCHet (Zaccaria, S. and Raphael, B.J., 2020, <https://doi.org/10.1038/s41467-020-17967-y>)). Additionally, robust benchmarking of such analyses would involve an extensive piece of work and we therefore intend to address this in a future manuscript.

3. Assess mutation clustering

In addition to comparing inferred CCF to ground-truth CCF, it would be good to compare mutation clustering accuracy using V-measure or (adjusted) Rand index.

We calculated the mean absolute (some samples resulted in small negative values) adjusted Rand index for each pipeline. These closely mirrored the results from when comparing CCF accuracy.

Line 293: "Additionally, mean absolute adjusted Rand index is calculated to indicate the accuracy of the variant clustering in each pipeline (Supplemental Table 5). When using non-clustered CCF estimates the best performing pipeline is FACETS with PyClone, though this only marginally improves upon the baseline of doubled VAFs and only at sequencing depths $\geq 100x$ (Figure 5a). When using clustered CCF estimates, the MADif and adjusted Rand index metrics largely agreed, with the best performing pipeline being FACETS with PyClone-VI run with beta binomial used for clustering."

Minor

* Page 3: "This was also used to incorporate immune cell infiltration to assess the effect of tumour purity at 75% vs 100%."

Immune cells are not the only type of normal cells.

We have corrected this error by replacing “immune” with “normal”.

Line 125: “This was also used to incorporate normal cell infiltration to assess the effect of tumour purity across a range of 25% to 100%.”

* Figure 3: Is panel (b) showing recall split by VAF? If so, change y-label to "Recall" as in panel (a). Also, it would be good to show precision split by VAF

The figure has been adapted as suggested.

* SNV clusters are not clones

Methods for SNV clustering do not infer tumor clones. Tumor clones can only be inferred in the context of a phylogenetic tree.

We have adjusted the wording throughout the manuscript to clarify that we do not infer clones. The main example is on line 374:

Original: “This enabled us to carry out reliable assessment of all steps needed for subclonal deconvolution pipelines, from unprocessed reads through somatic point and copy number mutation calling and through to subclone identification.”

->

“This enabled us to carry out reliable assessment of all steps from unprocessed reads through somatic point and copy number mutation calling and through to CCF inference and clustering. We did not attempt to assess methods for phylogenetic reconstruction as often multiple trees are outputted requiring manual curation, and which are not possible to categorically compare. Nonetheless, the accuracy of phylogenies will be heavily impacted by that of the inputted mutation CCF estimates.”

Reviewer #2 (Remarks to the Author): Expert in cancer genomics and evolution, clonal deconvolution, and bioinformatics

This is a study that benchmarks different tools for single nucleotide variant (SNV) calling, copy number aberration (CNA) analysis and subclonal deconvolution with the aim of finding the best combination to study tumor evolution from bulk whole-exome sequencing (WES) data. The authors first used their published tool to simulate tumor genomes with different subclonal structure as well as SNV and CNA rates. They then used another tool to generate whole exome sequencing data at a range of sequencing coverage (30x, 60x, 100x and 250x) and tumor purity (75% and 100%). This data is used to find the best combination from a set of four SNV callers, four CNA callers and five subclonal deconvolution tools. Their analysis indicates that Mutect2/FACETS/PyClone-VI combination gives the optimal solution. This is an interesting study but there are some important points that need to be addressed:

1. The choice of WES as the sequencing data is intriguing as there might not be enough mutations to cluster unless the tumor is a hypermutant (~10 mutations/Mb). A recent study showed that most studied tumors do not have a high mutation burden [Campbell et al Cell 2018]. 17% of the tumors had 10Muts/Mb and only 0.6% had 100Muts/Mb. These correspond to parameter set-2 and -3 in Figure 1. This means that only ~33% of their input data will reflect the mutation burden observed in

the majority of tumors. This could be exacerbated by using whole genome data. Did the authors check how the choice of WGS and WES would affect their results?

We thank the reviewer for these comments. We created WGS datasets using ReSeq's standard approach, for the best and worst performing tumours in the WES analysis, which were set 1 and set 2 tumours. The performance of the pipelines on these datasets was comparable to when using WES and did not alter our conclusions about the optimal methods to use. This also provides support that higher mutation numbers (from WGS vs WES) do not substantially impact pipeline performance, in agreement with our comparison of performance between the simulated tumour sets in Supplemental Figure 3.

Line 332: "To further assess the robustness of our findings, we ran the pipelines on additional WES datasets for samples with reduced numbers of clones and altered clone topologies, as well as the WGS datasets for the top (S1R1) and worst (S2R2) performing WES samples. In all investigations, the relative performance of pipelines was largely preserved, with FACETS with PyClone-VI_beta-binomial remaining the top performers (Supplemental Figures 4-6)."

Line 475: "WGS reads were created using Reseq's illuminaPE command."

Line 495: "WGS variant calls were limited to those outside of blacklist regions, from <https://github.com/Boyle-Lab/Blacklist/raw/master/lists/hg38-blacklist.v2.bed.gz>."

2. There are only two purity values taken into account: 75% and 100%. The authors should include lower purity values as 75% is still very high for most solid tumors.

We have now included 25% and 50% purities for the three tumours that previously resulted in the best performance. This showed that performance drops with decreasing purity but that the relative performance of pipelines generally remains the same.

Line 214: "We applied four somatic allele specific CNA callers (Sclust¹⁵, FACETS¹⁶, Sequenza¹⁷ and TITAN¹⁸) to samples from all nine simulated tumours at 100% and 75% purity and across sequencing depths of 30x-250x. We further included 50% and 25% purity samples at 250x for three of the tumours. Ploidy and purity estimates were largely accurate, particularly at $\geq 50\%$ purity, with the exceptions that TITAN often incorrectly assessed purity and Sequenza frequently overestimated ploidy (Figure 4, Supplemental Table 3)."

Line 268: "The performance of five subclonal deconvolution methods (PyClone¹⁹, PyClone-VI⁷, FastClone²⁰, Ccube²¹ and Sclust¹⁵) was assessed on all tumours and sequencing depths, at 100% and 75% purity."

Line 328: "We next sought to investigate the effects of lower tumour purities on the top three performing tumours (S1R1, S2R1, S2R3), using samples at 100%, 75%, 50% and 25% purity. This showed a decrease in accuracy with decreasing purity, particularly at 25% (Supplemental Figure 4), which likely partially results from the poorer purity estimates from CNA callers at that purity....
...In all investigations, the relative performance of pipelines was largely preserved, with FACETS with PyClone-VI_beta-binomial remaining the top performers (Supplemental figures 4-6)."

3. I am a bit surprised that Mutect2 came out as the caller of choice. Our experience with Mutect2 using WGS is that it has a higher false positive rate compared to Strelka2 especially in the low VAF range. That is the reason why we always prefer ensemble calling to using a single mutation caller. But the difference may be because the use of WES in this study.

This is an interesting point. We repeated our variant calling investigation using both WGS and WES at 100x coverage of S2R2 and still found that Mutect2 (after applying FilterMutectCalls) has a lower false positive rate than Strelka2.

Line 199: “To investigate potential differences between WES and whole genome sequencing (WGS), we created *in silico* WGS reads for sample S2R2_B_100x and compared variant calling performance from this with WES for the same sample. Of note, Strelka2_filtered dropped in both precision and recall for WGS compared to WES, whereas Mutect2_filtered maintained a similar performance (Supplemental Table 3).”

4. The authors assessed the performance of subclonal deconvolution tools comparing 'true variant CCFs to the estimated non-clustered CCFs and the CCF of the clusters the variants are assigned to'. True variant CCFs are calculated as multiplying the VAF by two and dividing by purity. This does not take CN changes in account. Would the results be the same if the authors excluded mutations in genomic regions affected by CNA?

This was not the case and we apologise for the confusion. The doubled VAFs approach is meant as an overly simplistic alternative to applying a proper CCF estimation method, and which will inevitably be erroneous due the CNA's as the reviewer pointed out. The CCF estimation methods should therefore be expected at a very minimum to reach and improve on the performance of the doubled VAFs approach in order to justify their use.

Changes in bold:

Line 277: “The performance of each pipeline was determined through comparison of true variant CCFs (**calculated by summing the tumour proportions of clones that contained each variant**) to both the estimated non-clustered variant CCFs and the CCF of clusters that the variants were subsequently assigned to, where applicable.

...

To provide **additional** simplistic references upon which CCF estimation methods should improve, we developed baseline datasets via two approaches....”

Reviewer #1 (Remarks to the Author): Expert in intratumour heterogeneity, clonal deconvolution methods, and bioinformatics

This paper benchmarks several methods for deconvolution of bulk DNA-seq data of tumors. Specifically, the paper focuses on methods for SNV calling, CNA calling and SNV clustering. The paper is well written. I do have some comments regarding the novelty of the analysis.

1. Not the first benchmarking study

Paper states:

"We have, therefore, performed the most comprehensive, robust and accurate assessment of subclonal deconvolution analysis pipelines to date and have made our series of highly realistic tumour WES datasets, spanning a range of biological and dataset-driven parameters, available to the wider community for future benchmarking studies as new tools emerge."

A similar study appeared recently, also in Nature Comm:

Liu, L. Y., Bhandari, V., Salcedo, A., Espiritu, S. M. G., Morris, Q. D., Kislinger, T., & Boutros, P. C. (2020). Quantifying the influence of mutation detection on tumour subclonal reconstruction. Nature Communications, 11(1), 6247. <https://doi.org/10.1038/s41467-020-20055-w>

This decreases the novelty of the paper.

We thank the reviewer for pointing out this study which we have now mentioned in our introduction. Unlike our approach, Liu et al. do not compare pipeline outputs to known ground truths and instead compare pipelines run on real data, allowing for interesting observations on biases but without estimation of the overall accuracy of methods. There was also a need for assessment of more up to date variant callers (eg. Liu et al., 2020 use Mutect1 v1.1.4, whereas the latest version is Mutect2 v4.2.0). We therefore believe our study compliments results from the Liu et al. study but has important distinctions that allow it to be considered novel.

Line 107: "Other studies have used real data to draw comparisons between pipelines, allowing for biases to be identified but with no certainty of the ground truths³."

2. Simulations

Abstract states: "... the most comprehensive tumour genome simulation tool available ...".

There is another tool for doing this: MASCoTE, published in:

Zaccaria, S., & Raphael, B. J. (2020). Accurate quantification of copy-number aberrations and whole-genome duplications in multi-sample tumor sequencing data. Nature Communications, 11(1), 4301. <https://doi.org/10.1038/s41467-020-17967-y>

We also thank the reviewer for pointing out this study. MASCoTE simulates copy number alterations but does not include simulation of SNVs or small InDels and is therefore not suited to the analyses of the current study. Given the tight word limit of the manuscript it is not possible to go into details of individual alternative simulation tools but we hope our slight change to the wording better implies that HeteroGenesis is the most comprehensive out of those suitable for the purposes of our study.

Line 16: "... but there is a lack of systematic benchmarking to inform researchers on which are most accurate, and how dataset characteristics impact performance. To address this, we used the most comprehensive tumour genome simulation tool available **for such purposes** to create 80 bulk tumour whole exome sequencing datasets of differing depths, tumour complexities, and purities, and used these to benchmark subclonal deconvolution pipelines."

In addition, it would be good to vary the tree topology as well as the number of clones. Also, varying the number of samples would be good to do.

We agree that the study would benefit from additional variations of the samples. To address this we created additional samples with 3 and 5 clones, and which have a linear or branched tree topology respectively, and ran the pipelines on these. The relative performance of the different pipelines was largely unchanged and did not alter our conclusions.

Line 332:" To further assess the robustness of our findings, we ran the pipelines on additional WES datasets for samples with reduced numbers of clones and altered clone topologies, as well as the WGS datasets for the top (S1R1) and worst (S2R2) performing WES samples. In all investigations, the relative performance of pipelines was largely preserved, with FACETS with PyClone-VI_beta-binomial remaining the top performers (Supplemental Figures 4-6)."

The suggestion of varying the number of samples is also of interest. However we believe a multisample set-up to be a distinct scenario (for example it would benefit from including variant and copy number callers specifically designed to be used with multiple samples such as HATCHet (Zaccaria, S. and Raphael, B.J., 2020, <https://doi.org/10.1038/s41467-020-17967-y>)). Additionally, robust benchmarking of such analyses would involve an extensive piece of work and we therefore intend to address this in a future manuscript.

3. Assess mutation clustering

In addition to comparing inferred CCF to ground-truth CCF, it would be good to compare mutation clustering accuracy using V-measure or (adjusted) Rand index.

We calculated the mean absolute (some samples resulted in small negative values) adjusted Rand index for each pipeline. These closely mirrored the results from when comparing CCF accuracy.

Line 293: "Additionally, mean absolute adjusted Rand index is calculated to indicate the accuracy of the variant clustering in each pipeline (Supplemental Table 5). When using non-clustered CCF estimates the best performing pipeline is FACETS with PyClone, though this only marginally improves upon the baseline of doubled VAFs and only at sequencing depths $\geq 100x$ (Figure 5a). When using clustered CCF estimates, the MADif and adjusted Rand index metrics largely agreed, with the best performing pipeline being FACETS with PyClone-VI run with beta binomial used for clustering."

Minor

* Page 3: "This was also used to incorporate immune cell infiltration to assess the effect of tumour purity at 75% vs 100%."

Immune cells are not the only type of normal cells.

We have corrected this error by replacing “immune” with “normal”.

Line 125: “This was also used to incorporate normal cell infiltration to assess the effect of tumour purity across a range of 25% to 100%.”

* Figure 3: Is panel (b) showing recall split by VAF? If so, change y-label to "Recall" as in panel (a). Also, it would be good to show precision split by VAF

The figure has been adapted as suggested.

* SNV clusters are not clones

Methods for SNV clustering do not infer tumor clones. Tumor clones can only be inferred in the context of a phylogenetic tree.

We have adjusted the wording throughout the manuscript to clarify that we do not infer clones. The main example is on line 374:

Original: “This enabled us to carry out reliable assessment of all steps needed for subclonal deconvolution pipelines, from unprocessed reads through somatic point and copy number mutation calling and through to subclone identification.”

->

“This enabled us to carry out reliable assessment of all steps from unprocessed reads through somatic point and copy number mutation calling and through to CCF inference and clustering. We did not attempt to assess methods for phylogenetic reconstruction as often multiple trees are outputted requiring manual curation, and which are not possible to categorically compare. Nonetheless, the accuracy of phylogenies will be heavily impacted by that of the inputted mutation CCF estimates.”

Reviewer #2 (Remarks to the Author): Expert in cancer genomics and evolution, clonal deconvolution, and bioinformatics

This is a study that benchmarks different tools for single nucleotide variant (SNV) calling, copy number aberration (CNA) analysis and subclonal deconvolution with the aim of finding the best combination to study tumor evolution from bulk whole-exome sequencing (WES) data. The authors first used their published tool to simulate tumor genomes with different subclonal structure as well as SNV and CNA rates. They then used another tool to generate whole exome sequencing data at a range of sequencing coverage (30x, 60x, 100x and 250x) and tumor purity (75% and 100%). This data is used to find the best combination from a set of four SNV callers, four CNA callers and five subclonal deconvolution tools. Their analysis indicates that Mutect2/FACETS/PyClone-VI combination gives the optimal solution. This is an interesting study but there are some important points that need to be addressed:

1. The choice of WES as the sequencing data is intriguing as there might not be enough mutations to cluster unless the tumor is a hypermutant (~10 mutations/Mb). A recent study showed that most studied tumors do not have a high mutation burden [Campbell et al Cell 2018]. 17% of the tumors had 10Muts/Mb and only 0.6% had 100Muts/Mb. These correspond to parameter set-2 and -3 in Figure 1. This means that only ~33% of their input data will reflect the mutation burden observed in

the majority of tumors. This could be exacerbated by using whole genome data. Did the authors check how the choice of WGS and WES would affect their results?

We thank the reviewer for these comments. We created WGS datasets using ReSeq's standard approach, for the best and worst performing tumours in the WES analysis, which were set 1 and set 2 tumours. The performance of the pipelines on these datasets was comparable to when using WES and did not alter our conclusions about the optimal methods to use. This also provides support that higher mutation numbers (from WGS vs WES) do not substantially impact pipeline performance, in agreement with our comparison of performance between the simulated tumour sets in Supplemental Figure 3.

Line 332: "To further assess the robustness of our findings, we ran the pipelines on additional WES datasets for samples with reduced numbers of clones and altered clone topologies, as well as the WGS datasets for the top (S1R1) and worst (S2R2) performing WES samples. In all investigations, the relative performance of pipelines was largely preserved, with FACETS with PyClone-VI_beta-binomial remaining the top performers (Supplemental Figures 4-6)."

Line 475: "WGS reads were created using Reseq's illuminaPE command."

Line 495: "WGS variant calls were limited to those outside of blacklist regions, from <https://github.com/Boyle-Lab/Blacklist/raw/master/lists/hg38-blacklist.v2.bed.gz>."

2. There are only two purity values taken into account: 75% and 100%. The authors should include lower purity values as 75% is still very high for most solid tumors.

We have now included 25% and 50% purities for the three tumours that previously resulted in the best performance. This showed that performance drops with decreasing purity but that the relative performance of pipelines generally remains the same.

Line 214: "We applied four somatic allele specific CNA callers (Sclust¹⁵, FACETS¹⁶, Sequenza¹⁷ and TITAN¹⁸) to samples from all nine simulated tumours at 100% and 75% purity and across sequencing depths of 30x-250x. We further included 50% and 25% purity samples at 250x for three of the tumours. Ploidy and purity estimates were largely accurate, particularly at $\geq 50\%$ purity, with the exceptions that TITAN often incorrectly assessed purity and Sequenza frequently overestimated ploidy (Figure 4, Supplemental Table 3)."

Line 268: "The performance of five subclonal deconvolution methods (PyClone¹⁹, PyClone-VI⁷, FastClone²⁰, Ccube²¹ and Sclust¹⁵) was assessed on all tumours and sequencing depths, at 100% and 75% purity."

Line 328: "We next sought to investigate the effects of lower tumour purities on the top three performing tumours (S1R1, S2R1, S2R3), using samples at 100%, 75%, 50% and 25% purity. This showed a decrease in accuracy with decreasing purity, particularly at 25% (Supplemental Figure 4), which likely partially results from the poorer purity estimates from CNA callers at that purity....
...In all investigations, the relative performance of pipelines was largely preserved, with FACETS with PyClone-VI_beta-binomial remaining the top performers (Supplemental figures 4-6)."

3. I am a bit surprised that Mutect2 came out as the caller of choice. Our experience with Mutect2 using WGS is that it has a higher false positive rate compared to Strelka2 especially in the low VAF range. That is the reason why we always prefer ensemble calling to using a single mutation caller. But the difference may be because the use of WES in this study.

This is an interesting point. We repeated our variant calling investigation using both WGS and WES at 100x coverage of S2R2 and still found that Mutect2 (after applying FilterMutectCalls) has a lower false positive rate than Strelka2.

Line 199: “To investigate potential differences between WES and whole genome sequencing (WGS), we created *in silico* WGS reads for sample S2R2_B_100x and compared variant calling performance from this with WES for the same sample. Of note, Strelka2_filtered dropped in both precision and recall for WGS compared to WES, whereas Mutect2_filtered maintained a similar performance (Supplemental Table 3).”

4. The authors assessed the performance of subclonal deconvolution tools comparing 'true variant CCFs to the estimated non-clustered CCFs and the CCF of the clusters the variants are assigned to'. True variant CCFs are calculated as multiplying the VAF by two and dividing by purity. This does not take CN changes in account. Would the results be the same if the authors excluded mutations in genomic regions affected by CNA?

This was not the case and we apologise for the confusion. The doubled VAFs approach is meant as an overly simplistic alternative to applying a proper CCF estimation method, and which will inevitably be erroneous due the CNA's as the reviewer pointed out. The CCF estimation methods should therefore be expected at a very minimum to reach and improve on the performance of the doubled VAFs approach in order to justify their use.

Changes in bold:

Line 277: “The performance of each pipeline was determined through comparison of true variant CCFs (**calculated by summing the tumour proportions of clones that contained each variant**) to both the estimated non-clustered variant CCFs and the CCF of clusters that the variants were subsequently assigned to, where applicable.

...

To provide **additional** simplistic references upon which CCF estimation methods should improve, we developed baseline datasets via two approaches....”